# Evaluation of B7-H3 Targeted Immunotherapy in a 3D Organoid Model of Craniopharyngioma

**DOI:** 10.3390/biom12121744

**Published:** 2022-11-24

**Authors:** Mei Tang, Caili Chen, Guoqing Wang, Yuelong Wang, Zongliang Zhang, Hexian Li, Qizhong Lu, Zeng Wang, Shasha Zhao, Chen Yang, Kunhong Zhong, Ruyuan Zhang, Liping Guo, Zhu Yuan, Chunlai Nie, Aiping Tong

**Affiliations:** 1State Key Laboratory of Biotherapy and Cancer Center, Research Unit of Gene and Immunotherapy, Chinese Academy of Medical Sciences, Collaborative Innovation Center of Biotherapy, West China Hospital, Sichuan University, Chengdu 610041, China; 2Department of Immunology, School of Basic Medical Sciences, Xinxiang Medical University, Xinxiang 453003, China; 3Department of Neurosurgery, West China Hospital, West China Medical School, Sichuan University, Chengdu 610041, China

**Keywords:** craniopharyngioma, B7-H3, organoid, immunotherapy, brain cancer

## Abstract

A craniopharyngioma (CP) is a rare epithelial tumor of the sellar and parasellar region. CPs are difficult to treat due to their anatomical proximity to critical nervous structures, which limits the ability of the surgeon to completely resect the lesion, exposing patients to a high risk of recurrence. The treatment of craniopharyngiomas is primarily surgery and radiotherapy. So far, neither a cell line nor an animal model has been established, and thus data on other treatment options, such as chemotherapy and immunotherapy, are limited. Here, the expression profile of the pan-cancer antigen B7-H3 in various cancer types including CP was examined by immunohistochemistry. An in vitro organoid model was established by using fresh tissue biospecimens of CP. Based on the organoid model, we evaluated the antitumor efficacy of B7-H3-targeted immunotherapy on CP. As a result, the highest expression of B7-H3 was observed in CP tissues across various cancer types. Although B7-H3-targeted chimeric antigen-receptor T cells show obvious tumor-killing effects in the traditional 2D cell culture model, limited antitumor effects were observed in the 3D organoid model. The B7-H3-targeted antibody-DM1 conjugate exhibited a potent tumor suppression function both in 2D and 3D models. In conclusion, for the first time, we established an organoid model for CP and our results support that B7-H3 might serve as a promising target for antibody-drug conjugate therapy against craniopharyngioma.

## 1. Introduction

Craniopharyngioma (CP) is a slow-growing, benign, epithelial neoplasm in the sellar and parasellar region [1,2]. The WHO classified CPs into two types: adamantinomatous CP (ACP) and papillary CP (PCP), 30–50% of which are most commonly seen in children [3,4]. Although the 20-year overall survival rate in childhood-onset CP was 87–95%, the mortality rate of adult-onset CP patients is up to 19 times that of cerebrovascular mortality, and the recurrences and progressions are frequent [5]. The quality of life with CP is frequently impaired owing to the adverse effects, including focal neurological deficits, hypothalamic dysfunction, endocrinopathies, obesity and eating disorders, and ophthalmologic disturbances [1,2,3,4,5].

The treatment of CP is primarily surgery and radiotherapy [1,2]. In last two decades, the mutations in the genes encoding β-catenin (CTNNB1) and B-Raf (BRAF), which are considered as the main oncogenic drivers of ACP and PCP, have received enthusiastic attention [6,7]. The use of BRAF inhibitors in BRAFV600E mutant PCP has provided excellent results in tumor suppression, but the efficient inhibitor for CTNNB1 mutation is not yet available, and rapid CP regrowth has also been reported after BRAF inhibitor treatment [8,9,10]. So far, neither a cell line nor an animal model has been established, and thus, data on other treatment options, such as chemotherapy and immunotherapy, are limited.

Previously, B7-H3 (CD276) was reported to be highly expressed in both ACP and PCP [11]. B7-H3 is a novel immune checkpoint molecule highly expressed in many malignant tumors, whereas it is low in normal tissues. This molecule has immune-inhibitory functions [12], plays important roles in tumor immune evasion and metastasis, proliferation, migration, invasion, development of cancer stem cell enrichment, and drug resistance [13,14], and the overexpression of B7-H3 frequently correlates with fewer tumor-infiltrating lymphocytes and poor clinical outcome in several malignancies [15].

Several B7-H3-targeted therapies, such as chimeric antigen receptor (CAR)-T cells, antibody drug conjugates (ADC), and radionuclide drug conjugates (RDC), have entered into clinical trials. ^131^I-Omburtamab is currently in the most advanced stages of approval for neuroblastoma [16]. A phase 2 clinical trial with DS-7300A, a B7-H3 antibody conjugated with topoisomerase I inhibitor Dxd, has been launched by Daiichi Sankyo recently. There are also several clinical trials for CAR-T cell adoptive therapies targeting B7-H3: NCT04432649 sponsored by Shenzhen Geno-Immune Medical Institute, and NCT04483778 and NCT04185038 sponsored by Seattle Children’s Hospital.

In the present study, after the examination of B7-H3 expression profile across various cancer types including CP, we established for the first time an in vitro organoid model of CP using fresh tissue biospecimens, and based on the model, we evaluated the antitumor efficacy of B7-H3-targeted CAR-T cells and ADC against craniopharyngioma.

## 2. Materials and Methods

### 2.1. Clinical Samples

Patient samples were obtained under a West China Hospital approved protocol. Informed consent was obtained from all patients in accordance to the Declaration of Helsinki.

### 2.2. Immunohistochemistry

Clinical tumor tissue samples and commercially available tumor tissue chips were stained for B7-H3 using a commercial mAb (#14058S; Cell Signaling Technology) according to the manufacturer’s standard protocol. The immunohistochemical score was determined according to previous publications [11]. Briefly, the percentage of positive staining cells and cell staining intensity in five randomly selected fields were counted. The staining intensity was scored as follows: 0 (negative); 1 (weakly positive); 2 (moderately positive); and 3 (strongly positive). The overall score was quantified as: histopathological score = (1 × weak positive staining % + 2 × positive staining % + 3 × strong positive staining %) × 100.

### 2.3. Cell Culture and Lentivirus Packaging

The HEK293T, Hela, DU145, and A375 cell lines were originally obtained from American Type Culture Collection (ATCC). Hela-B7-H3-EGFP, DU145B7-H3-KO, and A375 B7-H3-KO cells were a gift from Shasha Zhao [17]. The cells were cultured in DMEM or RPMI-1640 supplemented with 10% fetal bovine serum, 100 U/mL penicillin, and 100 mg/mL streptomycin at 37 °C with 5% CO2. Lentivirus packaging was performed using HEK293T cells transfected with target plasmid, psPAX2, and pMD.2G at a ratio of 3:2:1 using polyethylenimine. Forty-eight hours post transfection, the supernatants were collected, filtered with a 0.45 μm filter, and further concentrated by ultracentrifugation at 100,000× g for 2 h. The concentrated lentivirus was resuspended in PBS and stored at −80 °C until use.

### 2.4. Monoclonal Antibody Generation, Affinity, and Epitope Determination

The generation of B7-H3-specific monoclonal antibodies and scFv as well as the affinity determination by Biacore analysis were performed using traditional hybridoma technology according to previously published methods [18]. Briefly, the extracellular domain of B7-H3 with C-terminal His-tag was transiently expressed in CHO cells by electroporation transfection and purified by nickel columns. After immunization of BALB/c mouse by B7-H3 recombinant protein, the splenocytes were isolated and fused with myeloma cells polyethylene glycol (PEG, Sigma, P7181). Ten days after fusion, the hybridoma cells were screened for the positive clones by ELISA, immunofluorescence, and flow cytometry, respectively. For epitope determination, six different extracellular domains (V1, C1, V2, C2, V1-C1, V2-C2) were cloned into an EGFP-tagged lentivirus vector and transfected mouse cell line MC38. After selection with puromycin, these stable cell lines were subjected to immunofluorescence staining with the B7-H3-specific monoclonal antibodies.

### 2.5. Immunofluorescence Staining and Flow Cytometry

For immunofluorescence staining, the cells were stained with primary antibody for 1 h at 4 °C, and then stained with fluorescein isothiocyanate (FITC)- or Cy3-conjugated secondary antibody (Proteintech, Chicago, IL, USA) and DAPI (Beyotime, Beijing, China). For the flow cytometry, the expression of CAR was examined by staining with a FITC-labeled recombinant protein of B7-H3 extracellular domain, and the BV510-labeled antiCD4 antibody (Biolegend, San Diego, CA, USA, 357420) and BV650-labled antiCD8 antibody (Biolegend, 344730) were used to determine the expression and ratio of CD8:CD4. Flow cytometry analysis was performed on a BD Fortessa flow cytometer and analyzed using FlowJo 10.6.0 software (BD, Franklin Lakes, NJ, USA).

### 2.6. B7-H3 CAR-T Cell Generation

The CAR-T cells were generated by the transfection of primary T cells with a lentivirus vector carrying intracellular activation domain of CD28/CD3 according to a previously published paper [19]. The scFv of antiB7-H3 was derived from monoclonal antibody 14A2. Briefly, peripheral blood PBMC cells from healthy volunteers were isolated using gradient centrifugation (800× *g*, 20 min). The harvested cells were cultured with 15% fetal calf serum, 100 U/mL penicillin, and 100 μg/mL streptomycin at 37 °C with 5% CO_2_. Then, the T cells were added with OKT3 (600 ng/mL, Novoprotein, Suzhou, China) and IL-7/IL15 co-stimulators (IL7, 10 ng/mL; IL15, 5 ng/mL; PeproTech, Suzhou, China). After 48 h, the activated T cells were transducted with lentivirus vector by low-speed centrifugation transfection and RetroNectin (Takara-Bio, Kyoto, Japan). After 7–10 days of amplification, the CAR-T cells were subjected to examination and functional research.

### 2.7. Preparation of B7-H3-Targeted Antibody Conjugate

The preparation of antibody-drug conjugate was performed according to the published literature [20] with minor modifications. Briefly, B7-H3 antibody 14A2 and DM1-SMCC were mixed in a molar ratio of 1:9 in conjugation buffer (50 mM potassium phosphate, 50 mM sodium chloride, 2 mM EDTA, pH 7.2) and stirred at room temperature for 8 h. After being centrifuged with an ultrafiltration tube, the 14A2-DM1 conjugates were replaced into the storage buffer (50 mM sodium phosphate, 50 mM sodium chloride, pH 7.2) (Beyotime, Shanghai, China). The drug-to-antibody ratios (DARs) of ADC were determined by the absorbance of 14A2-DM1 conjugates examined by a microplate reader at 280 and 252 nm, respectively. The DAR of all batches of antibody conjugation drugs in this study ranged from 2.4 to 3.2.

### 2.8. Three-Dimensional (3D) Culture of CP Organoids

Due to the higher expression of B7-H3 in PCP than ACP, the experiments were investigated based on the samples from PCP. The CP organoids were cultured as described previously [21]. At first, the cells isolated from primary CP tissues were suspended with Corning Matrigel (Corning, Cat# 356255, NY, USA) mixed with organoid medium. Then, 3D organoids were prepared by seeding CP tumor cells in ultralow attachment dishes (Corning) and grown in DMEM/F12 medium (Gibco) supplemented with 1% hormone mixture B27 (Thermo Fisher Scientific, Cat# 17504044, Shanghai, China), 10 ng /mL human recombinant epidermal growth factor EGF (Sino Biological, Cat# 10605-HNAE, Beijing, China), and 10 ng /mL human recombinant fibroblast growth factor FGF (Sino Biological, Cat# 10573-HNAE, Beijing, China) for one month. Subsequently, the tumor organoids were transferred into a 24-well plate for co-culture assay. Primary antibodies against B7-H3 (#14058S; Cell Signaling Technology, Boston, Massachusetts, USA), CD133 (ab278053, abcam, Cambridge, UK), CK-7 (ab181598, abcam), and CTNNB1 (ab237984, abcam) were used in IHC staining.

### 2.9. In Vitro Tumor Killing and Inhibition Assays

The 2D cells and 3D organoids were co-cultured with B7-H3 CAR-T cells in 24-well plates. In the 2D cell model, the tumor cells were co-cultured with antiB7-H3 CAR-T cells at an E:T ratio of 4:1, and monitored by the xCELLigence real-time cell analyzer (ACEA Biosciences, San Diego, CA, US). In the 3D models, the CP organoids were initially transfected with lentivirus carrying mCherry, and the antiB7-H3 CAR-T cells were used at a number of 2 × 10^4^/wells after staining with CFSE (Biolegend, Cat# 423801, San Diego, CA, US). For ADC treatment, 2D and 3D cells were treated with antiB7-H3-DM1 at 5 nM. Images were captured at 24 h after co-culturing by a confocal microscope (Zeiss 880, Oberkochen, Germany). The diameter of the tumor organoids can be calculated by manual measurement under a light microscope assisted with computer-based imaging software.

### 2.10. Statistics

The data are expressed as mean ± SD. Two-tailed Student’s t tests were performed using Microsoft Excel and GraphPad Prism 8. *p* < 0.05 was considered to be significant, and the significance levels are represented as follows: ∗, *p* < 0.05; ∗∗, *p* < 0.01; ∗∗∗, *p* < 0.001.

## 3. Results

### 3.1. IHC Staining Analysis of B7-H3 in Clinical Tumor Tissue Samples

The expression of B7-H3 was examined by IHC across 12 types of cancer. As shown in Figure 1, B7-H3 was positively detected in all of the examined tumor tissue samples. A higher expression of B7-H3 was observed in head and neck squamous cell carcinoma (HNSCC), CP, prostate adenocarcinoma (PRAD), and glioblastoma (GBM), and among these, CPs show particularly high and homogeneous expression. Low and heterogeneous expressions of B7-H3 were detected in stomach adenocarcinoma (STAD), pancreatic adenocarcinoma (PAAD), kidney renal clear cell carcinoma (KIRC), and lung adenocarcinoma (LUAD), and median expression levels were observed in the remaining four types of cancer (cutaneous squamous cell carcinoma (CSCC), rectum adenocarcinoma (READ), ductal carcinoma in situ (DCIS), and hepatocellular carcinoma (HCC)).

### 3.2. Generation and Characterization of B7-H3-Specific Monoclonal Antibody and scFv

After purification by protein A, full antibody 14A2 and the scFv-Fc were analyzed by SDS-PAGE electrophoresis (Figure 2A). The specificity of 14A2 was validated by immunofluorescence staining using the Hela-B7-H3-EGFP cell line. As shown in Figure 2B, both the full antibody and the scFv-Fc can bind to B7-H3-positive cells. We also detected the binding of 14A2 antibodies on the B7-H3-positive native cell line, including NIH1975, DU145, and Hela (Appendix A). We further verified the specificity of 14A2 using a B7-H3 knockout DU145 cell line (Figure 2C). The confirmation of the knockout of B7-H3 in DU145 cells by FACS is shown in Appendix A. Figure 2D shows the positive IHC staining of CP tissue using the 14A2 antibody. The KD values of 14A2 full antibody and scFv-Fc binding to B7-H3 are 1.64 × 10^−9^ M and 4.04 × 10^−9^ M, respectively (Figure 2E). The binding epitope of 14A2 antibody was V1C1 and V2C2 (Appendix A).

### 3.3. Preparation of B7-H3 Targeted CAR-T Cells

Figure 3A shows the elements of the lentivirus vector encoding the B7-H3 CAR. The ScFv of antiB7-H3 was derived from clone 14A2. Forty-eight hours post stimulation, the T cells were transfected with lentivirus. After 7–10 days amplification, the CAR-T cells were subjected to FACS analysis. As shown in Figure 3B,C, the median value of transfection efficiency was 73%, which was determined by staining with FITC-labeled recombinant protein of B7-H3 extracellular domain. CD4-positive T cells accounted for about 21%, and the percentage of CD8-positive T cells was significantly higher, with an average value of over 50%. The ratio of CD8:CD4 was about 2.5 (Figure 3B,D). There was no significant difference in the ratio between control T cells and CAR-T cells stimulated and cultured under the same conditions (Figure 3E).

### 3.4. In Vitro Specific Antitumor Effects of B7-H3-Targeted CAR-T Cells and ADC

The specific antitumor effects of B7-H3-targeted CAR-T cells were evaluated with B7-H3 wild-type and knockout A375 cells. The confirmation of the knockout of B7-H3 in A375 cells by FACS is shown in Appendix A. As shown in Figure 4A, the B7-H3 wild-type cells were dramatically killed by the CAR-T cells, while there was no significant cell suppression in the knockout cells. Similarly, antiB7-H3-DM1 exhibited obvious cell-killing effects on wild-type cells with an IC50 at 5.4 nM, and no cell inhibition effects were observed in B7-H3 knockout cells (Figure 4B). We also detected the endocytosis of the 14A2 antibody, and as shown, the 14A2 antibodies could be internalized efficiently. After incubating at 37 °C for 2 h, 4 h, and 8 h, the internalization efficiency of the 14A2 antibody reached about 25%, 40%, and 60%, respectively, in A375 cells (Appendix A).

### 3.5. Establishment of Craniopharyngioma 3D Organoid Model

Images of CP organoids at different culture times are shown in Figure 5A. As shown, after 15 days of culture, the average diameter of the CP organoids could reach over 100 μm (Figure 5B). As shown in H&E staining, the histomorphologies of the cultured CP organoids and clinical tumor tissues are very similar (Figure 5C). The expression of B7-H3 and several known CP markers, including CD133, CK-7, and CTNNB1, were positively detected by IHC staining (Figure 5D).

### 3.6. Tumor Inhibition Evaluation with CP Organoid Model

The organoids from six B7-H3+ clinical CP samples were cultured successfully. Then, we evaluated the cell-killing effects of B7-H3-targeted CAR-T cells and ADC using these organoid models. Representative images and a statistical analysis of the tumor suppression effects are shown in Figure 6. Figure 6A shows the antitumor effects of B7-H3-targeted CAR-T cells on CP organoids. When the diameter of CP organoids is about 100 μm or beyond, only limited inhibition effects were observed after treatment with CAR-T cells (Figure 6B). In contrast, antiB7-H3-DM1 exhibited obvious tumor-killing effects on CP organoids from six clinical samples (Figure 6C,D).

## 4. Discussion

Craniopharyngiomas are classified into adamantinomatous craniopharyngioma (ACP) and papillary craniopharyngioma (PCP) subtypes. The primary treatments are surgery and radiotherapy, but the surgical resection of CPs is challenging, and recurrence is common. Genetic and immunological markers show variable expression levels in different types of CP [1,2]. It is known that BRAF V600E mutations are the main drivers of PCP, and thus PCP is sensitive to BRAF/MEK inhibitors [7]. However, effective targeted therapies are still needed for ACP. It was reported that CTNNB1 and EGFR are often overexpressed in ACP. Targeted treatment modalities inhibiting the CTNNB1 and EGFR pathways may be effective in the inhibition of ACP progression [6,22].

Immunotherapies, including IL-6 and interferon treatment, are effective in managing CP tumor growth [23]. The feasibility of targeting the programmed cell death protein 1/programmed death-ligand 1 (PD-1/PD-L1) immune checkpoint pathway in ACP and PCP has been discussed [24]. Recently, the expression of PD-L1, B7-H3, B7-H4, and VISTA was investigated in 132 CP cases using immunohistochemistry. The authors found the positive rates of PD-L1, B7-H3, B7-H4, and VISTA were 76.5%, 100%, 40.2%, and 80.3%, respectively, which suggests that these B7 family members could be potential immunotherapeutic targets against CPs [25]. In recent years, B7-H3 has attracted great attention given its prominent upregulation and immunomodulatory role in various cancer types. Several B7-H3-targeted therapies, including CAR-T cells, ADC, and radionuclide drug conjugates, have entered into clinical trials.

So far, neither a cell line nor an animal model has been established, which restricts the development of drugs against CP. In the present study, we established a CP organoid model for the first time, and using this model, we found that although B7-H3-targeted CAR-T cells show obvious tumor-killing effects in the traditional 2D cell culture model, limited antitumor effects were observed in the 3D organoid model, while the B7-H3-targeted antibody-DM1 conjugate exhibited a potent tumor suppression function in both the 2D and 3D models. CAR-T cells have demonstrated great success in treating hematological malignancies, but their activity in solid tumors has been unsatisfactory. The main obstacles to treating solid tumors include T cell trafficking and distribution throughout the tumor tissues, overcoming the immunosuppressive tumor microenvironment, and identifying homogeneous highly expressed targets [26]. Our results showed that the histomorphologies of the cultured CP organoids are very similar to clinical tumor tissues (Figure 5C). The poor infiltration ability of CAR-T cells into CP organoids may contribute to the compromised antitumor efficacy. Consistently, our data show that the killing effects of CAR-T on CP organoids were significantly compromised compared to the 2D co-cultures.

ADCs are composed of a monoclonal antibody, which specifically recognizes a cellular surface antigen linked to a cytotoxic payload [27]. ADCs have demonstrated superior efficacy and reduced toxicity in a range of hematological and solid tumors, resulting in twelve FDA and EMA approvals. In the present study, we profiled the expression of B7-H3 across twelve types of cancer and found that craniopharyngioma tissues show the highest expression. Using a 3D organoid model, we show that B7-H3-targeted antibody-DM1 exhibited more potent antitumor effects than the CAR-T cells. Recently, the topoisomerase I inhibitor Dxd-conjugated antibody against HER2 (DS-8201a) exhibited a superior antitumor activity in a broad selection of HER2-positive models, and it is suggested to be a valuable therapy with great potential to respond to T-DM1-insensitive HER2-positive cancers and low HER2-expressing cancers [28]. A phase 2 clinical trial with DS-7300A, a B7-H3 antibody conjugated with topoisomerase I inhibitor Dxd, has been launched by Daiichi Sankyo recently. Therefore, targeting B7-H3 by ADCs will be a very promising therapeutic strategy for CP.

## 5. Conclusions

We profiled the expression of B7-H3 across twelve types of cancer and found that craniopharyngioma tissues show the highest expression. For the first time, a CP organoid model was established, and using this model, our data demonstrated that although both B7-H3-targeted CAR-T cells and ADC possess obvious tumor-killing effects in the traditional 2D cell culture model, ADC exhibited a more potent tumor suppression function in the 3D models. Our data indicated that the targeting of B7-H3 by antibody–drug conjugates is a promising therapeutic strategy for the treatment of craniopharyngioma.

## Figures and Tables

**Figure 1 biomolecules-12-01744-f001:**
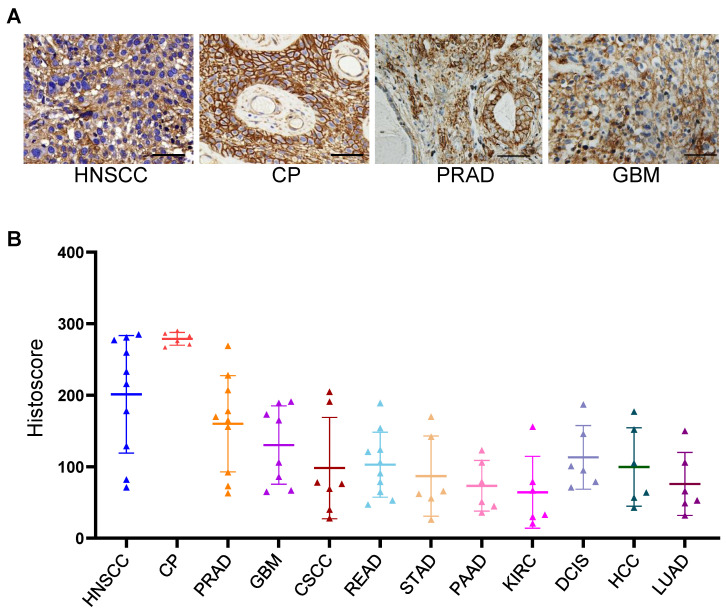
IHC staining analysis of B7-H3 expression in clinical tumor samples. (**A**) Immunohistochemical staining analysis. (**B**) Semi-quantitative analysis. Scale bar: 50 μm.

**Figure 2 biomolecules-12-01744-f002:**
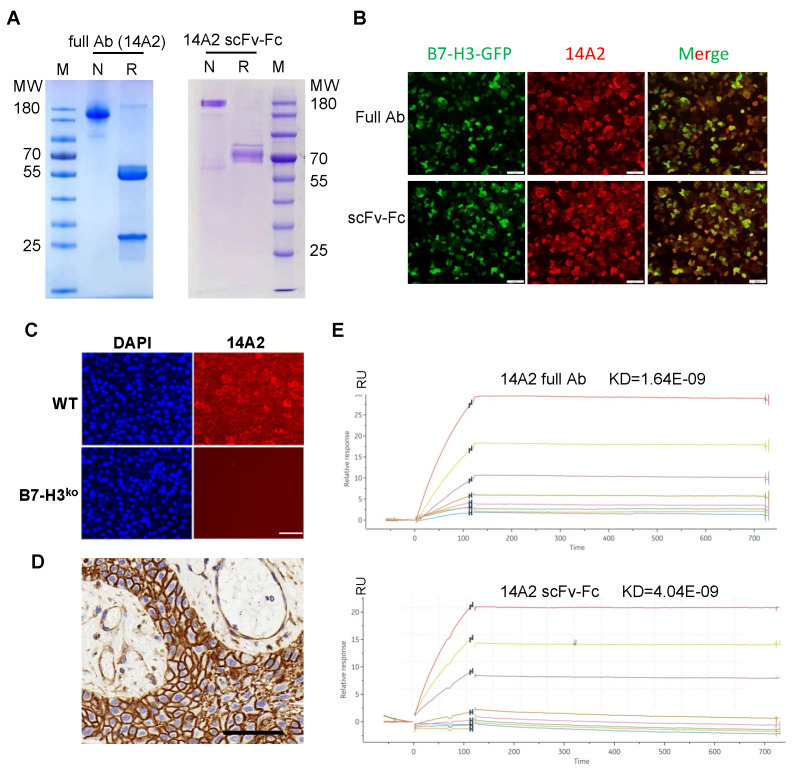
Validation and characterization of B7-H3-targeting monoclonal antibody and scFv. (**A**) SDS-PAGE analysis of purified antibody clone 14A2 and the derived scFv-Fc. (**B**) Immunofluorescence staining of 14A2 full antibody and scFv binding to Hela cells with B7-H3-EGFP stably expressing, scale bar: 100 μm. (**C**) Wild and B7-H3KO DU145 cells were used to verify the binding specificity of 14A2 by IF staining, scale bar: 100 μm. (**D**) Immunohistochemical staining analysis of CP clinical tumor samples with 14A2 antibody, scale bar: 50 μm. (**E**) Affinity analysis of 14A2 antibody and the derived scFv-Fc fragment to recombinant extracellular domain protein of B7-H3. R, reducing; N, non-reducing; M, marker.

**Figure 3 biomolecules-12-01744-f003:**
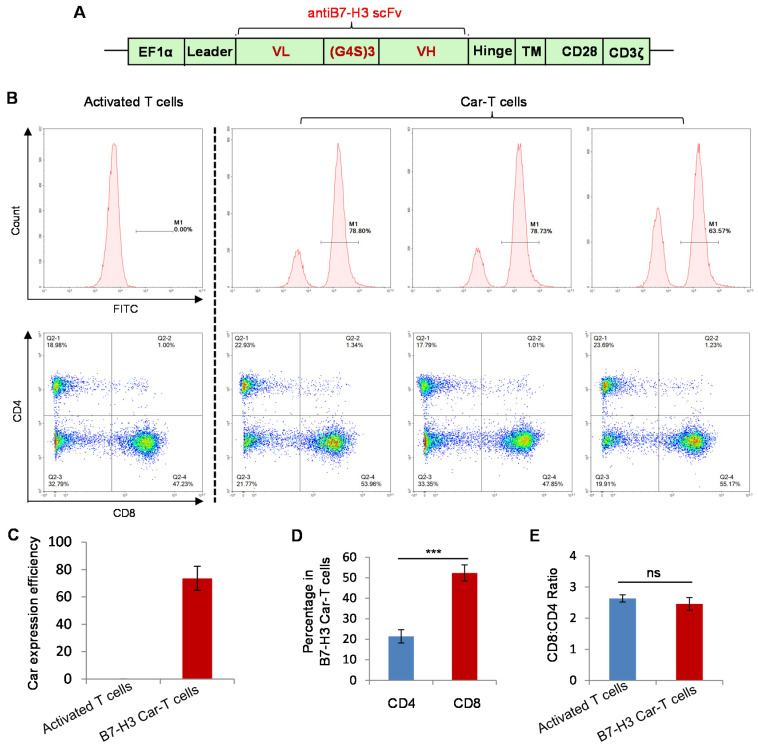
Preparation of B7-H3-targeted CAR-T cells. (**A**) Schematic diagram of the construction of the antiB7-H3 CAR lentiviral vector. (**B**–**E**) Detection of CAR expression after transfection of activated T cells by lentiviral vector, as well as the analysis of the ratio between CD4 and CD8. Statistical analysis was performed using two-tailed Student’s t test. Experiments were repeated at least three times, and data are expressed as mean ± standard deviation (mean ± SD). ***, *p* < 0.001; ns, not significant.

**Figure 4 biomolecules-12-01744-f004:**
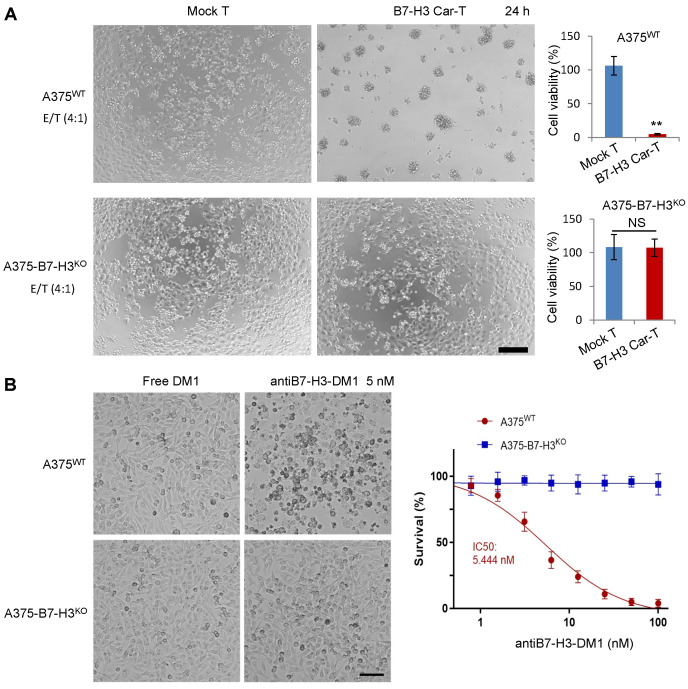
In vitro antitumor analysis of B7-H3-targeted CAR-T cells and ADC. (**A**) 0.5 × 10^4^/well tumor cells were plated in 96-well plates, and the ratio of effector to target (lymphocytes: tumor cells) was 4:1. The co-culture time was 24 h. Experiments were repeated at least three times, and data are expressed as mean ± standard deviation (mean ± SD). Statistical analysis was performed using two-tailed Student’s t test. **, *p* < 0.01; NS, not significant. Scale bar: 100 μm. (**B**) In vitro activity assay of B7-H3-targeting ADCs on A375 cells. After treatment with different concentrations of ADC drug for 72 h, cells were observed under an inverted microscope. The cell viability was determined using the CCK8 method, and the IC50 value was calculated with GraphPad 8.0. Scale bar: 100 μm. Experiments were repeated at least three times, and data are expressed as mean ± standard deviation (mean ± SD). Statistical analysis was performed using two-tailed Student’s t test.

**Figure 5 biomolecules-12-01744-f005:**
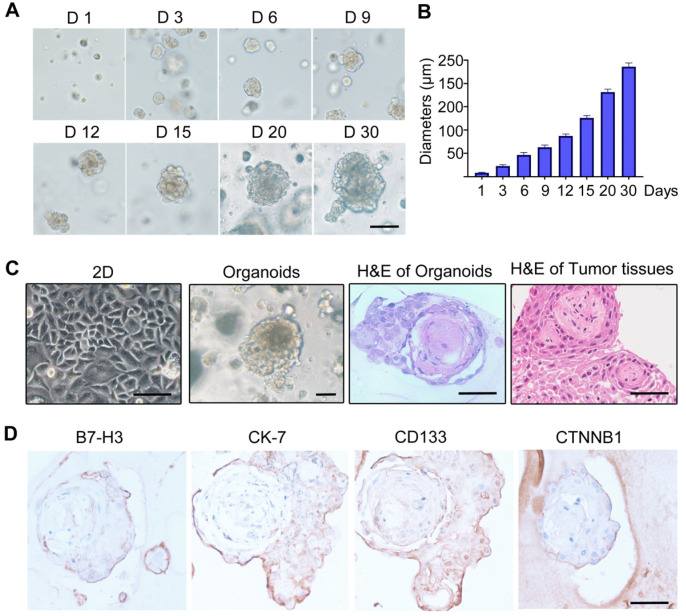
Establishment of 3D organoid in vitro cultures of craniopharyngioma. (**A**) Morphology of craniopharyngioma organoids at different time points. Scale bar: 100 μm. (**B**) Semi-quantitative analysis of organoid size. (**C**) Representative images of 2D culture of primary isolated CP cells, 3D organoids, and H&E staining of CP organoids and clinical tumor tissues. (**D**) IHC staining of B7-H3 and CP markers of the organoids. Scale bar: 100 μm.

**Figure 6 biomolecules-12-01744-f006:**
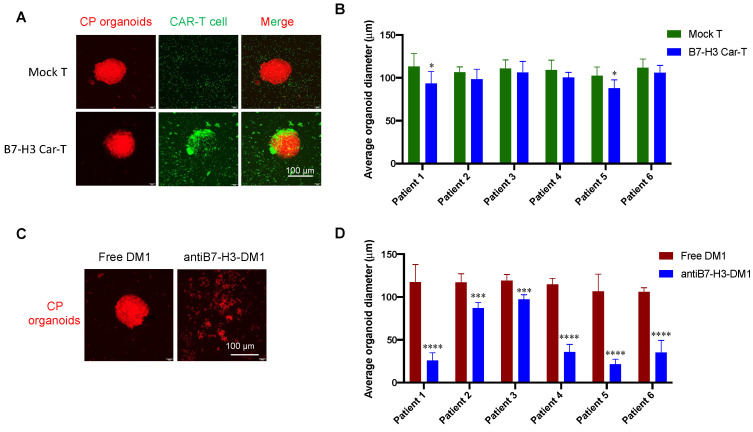
Evaluation of the antitumor activities of B7-H3-targeted CAR-T cells and ADC in CP organoid model. (**A**,**B**) Antitumor effects of B7-H3-CAR-T cells in CP organoid model from a representative sample. (**C**,**D**) Antitumor effects of antiB7-H3-DM1 ADC (5 nM) in CP organoid model. The CP organoids were initially transfected with lentivirus-carrying mCherry, and the antiB7-H3 CAR-T cells were used at a number of 2 × 10^4^/well after staining with CFSE. Images were captured at 24 h after co-culture or treatment with confocal microscope. Experiments were repeated at least three times, and data are expressed as mean ± standard deviation (mean ± SD). Statistical analysis was performed using two-tailed Student’s t test. *, *p* < 0.05; ***, *p* < 0.001; ****, *p* < 0.0001. Scale bar: 100 μm.

## Data Availability

The original contributions presented in the study are included in the article/supplementary material. Further enquiries can be directed to the corresponding authors.

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
