# Peer review of "Evaluation of B7-H3 Targeted Immunotherapy in a 3D Organoid Model of Craniopharyngioma"

_biomolecules, 2022, doi:10.3390/biom12121744_

Round 1
Reviewer 1 Report
The Manuscript titled as “Evaluation of B7-H3 targeted immunotherapy in a 3D organoid model of craniopharyngioma” first described an establishment on organoid model of craniopharyngioma (CP) using fresh tissue biospecimens, and the antitumor efficacy of B7-H3 targeted CAR-T cells and ADC against CP by using the organoid model. The manuscript also showed the expression profile of the pan-cancer antigen B7-H3 in various cancer types, especially in CP, and explored the anti-tumor potential of B7-H3 targeted antibody-DM1 conjugate on CP treatment.
The manuscript is significant in the field of immunotherapy for craniopharyngioma, and interesting to readers who are focusing on cancer treatment. However, there are several minor issues which need to be address prior to acceptance.
(1) These authors claimed that “Higher expression of B7-H3 was observed in HNSCC, CP, PRAD and GBM, and among these……..” However, in panel A of figure 1, B7-H3 expression in PRAD and GBM tissues was not seen.
(2) These authors used a lot of word abbreviations, especially for cancer types (for examples HNSCC, CP, PRAD and GBM… STAD, PAAD, KIRC and LUAD). Who knows what these abbreviations represented? Write out the full names when they are present in manuscript at the first time.
Author Response
Response letter
On behalf of the authors, I deeply appreciate your very careful review of our manuscript. They were very valuable and useful to improve our work. We have tried our best to address all the concerns raised by the reviewers and have revised our manuscript accordingly which we hope meet with approval. Our point-by-point responses to the reviewer's comments are presented below:
Response to comments from Reviewer #1
The Manuscript titled as “Evaluation of B7-H3 targeted immunotherapy in a 3D organoid model of craniopharyngioma” first described an establishment on organoid model of craniopharyngioma (CP) using fresh tissue biospecimens, and the antitumor efficacy of B7-H3 targeted CAR-T cells and ADC against CP by using the organoid model. The manuscript also showed the expression profile of the pan-cancer antigen B7-H3 in various cancer types, especially in CP, and explored the anti-tumor potential of B7-H3 targeted antibody-DM1 conjugate on CP treatment.
The manuscript is significant in the field of immunotherapy for craniopharyngioma, and interesting to readers who are focusing on cancer treatment. However, there are several minor issues which need to be address prior to acceptance.
- These authors claimed that “Higher expression of B7-H3 was observed in HNSCC, CP, PRAD and GBM, and among these……..” However, in panel A of figure 1, B7-H3 expression in PRAD and GBM tissues was not seen.
Answer: Thank you for your careful reading. After carefully check, we found that the Figure 1 in the uploaded Word file was correct, but the Figure 1 in the PDF file was wrong. It seems something went wrong when Word was converted to PDF. We have resubmitted the revised manuscript. If Figure 1 in the PDF file was still wrong, we can provide the original PPT file.
- These authors used a lot of word abbreviations, especially for cancer types (for examples HNSCC, CP, PRAD and GBM… STAD, PAAD, KIRC and LUAD). Who knows what these abbreviations represented? Write out the full names when they are present in manuscript at the first time.
Answer: We have added the full names of the abbreviations in the revised manuscript accordingly.

Reviewer 2 Report
In this study, Tang et al. reported that B7-H3 exhibited the highest expression level in craniopharyngioma across twelve types of cancer by IHC staining of clinical samples. So far, neither a cell line nor an animal model has been established for CPs. Thus, the authors established an organoid model for CP, and they demonstrated that B7-H3 targeted ADC exhibited potent tumor suppression function in both 2D and 3D cultures using this in vitro model, while anti-B7H3 CAR-T cells only possessed obvious tumor-killing effects in traditional 2D cultures, suggesting that targeting B7-H3 by ADC is a promising strategy for CP treatment.
In general, the manuscript is well written and their major conclusions are supported by the data presented. This work is interesting and highlights a potential therapeutic target for CP treatment. This manuscript would be more appropriate for publication with the following concerns addressed:
1) Craniopharyngiomas are rare malformational tumors of low histological malignancy arising along the craniopharyngeal duct. There are two histological subtypes, adamantinomatous craniopharyngioma (ACP) and papillary craniopharyngioma (PCP), differ in genesis and age distribution. So, what subtype was investigated in this study with regard to the IHC and organoid experiments?
2) The materials and methods are not descripted in detail. Please provide the detail materials and methods for these experiments, such as antibody and CAR-T cell generation.
3) Please proofread the manuscript carefully. Figure 2E was not cited in the main text. There are also some typographical and grammatical mistakes.
4) The author should add a discussion addressing why CAR-T was not efficient in the suppression of CP organoids.
5)How many clinical samples were used in this study for primary cell isolation and how about the success rate for the organoid culture?
6) In the present study, limited antitumor effects were observed in CAR-T cells against CP organoids. Did the author try CAR-T cells manufactured from different donors?
7)Were these samples all checked for B7-H3 expression and how about the positive rate?
Author Response
Response letter
On behalf of the authors, I deeply appreciate your very careful review of our manuscript. They were very valuable and useful to improve our work. We have tried our best to address all the concerns raised by the reviewers and have revised our manuscript accordingly which we hope meet with approval. Our point-by-point responses to the reviewer's comments are presented below:
Response to comments from Reviewer #2
In general, the manuscript is well written and their major conclusions are supported by the data presented. This work is interesting and highlights a potential therapeutic target for CP treatment. This manuscript would be more appropriate for publication with the following concerns addressed:
- Craniopharyngiomas are rare malformational tumors of low histological malignancy arising along the craniopharyngeal duct. There are two histological subtypes, adamantinomatous craniopharyngioma (ACP) and papillary craniopharyngioma (PCP), differ in genesis and age distribution. So, what subtype was investigated in this study with regard to the IHC and organoid experiments?
Answer: Thank you for your valuable comment. Due to the higher expression of B7-H3 in PCP than ACP, the IHC and organoid experiments were performed based on PCP samples. We have added this information in the Materials and Methods section of the revised manuscript.
- The materials and methods are not descripted in detail. Please provide the detail materials and methods for these experiments, such as antibody and CAR-T cell generation.
Answer: According to your suggestion, we have added the detail information regarding to antibody and CAR-T generation in the material and method section in the revised manuscript.
- Please proofread the manuscript carefully. Figure 2E was not cited in the main text. There are also some typographical and grammatical mistakes.
Answer: Thank you very much for your careful reading. We have cited Figure 2E and corrected the typographical and grammatical mistakes in the revised manuscript accordingly.
- The author should add a discussion addressing why CAR-T was not efficient in the suppression of CP organoids.
Answer: Thank you for your valuable comments. We have added a discussion addressing the reason of killing failure of Car-T cells against CP organoids in the revised manuscript.
- How many clinical samples were used in this study for primary cell isolation and how about the success rate for the organoid culture?
Answer: In this study, totally 6 clinical samples were used for establishing of CP organoid model. Based on our experience, CP organoids are easy to grow and passage in matrigel and organoid models were successfully established from all the six samples.
- In the present study, limited antitumor effects were observed in CAR-T cells against CP organoids. Did the author try CAR-T cells manufactured from different donors?
Answer: We have tried CAR-T cells from two different donors and similar results were obtained.
- Were these samples all checked for B7-H3 expression and how about the positive rate?
Answer: We checked all the 6 CP samples for B7-H3 expression, and the positive rate was 100%.
